# Altered Gut Microbiota and Its Clinical Relevance in Mild Cognitive Impairment and Alzheimer’s Disease: Shanghai Aging Study and Shanghai Memory Study

**DOI:** 10.3390/nu14193959

**Published:** 2022-09-23

**Authors:** Zheng Zhu, Xiaoxi Ma, Jie Wu, Zhenxu Xiao, Wanqing Wu, Saineng Ding, Li Zheng, Xiaoniu Liang, Jianfeng Luo, Ding Ding, Qianhua Zhao

**Affiliations:** 1Institute of Neurology, Huashan Hospital, Fudan University, Shanghai 200040, China; 2National Clinical Research Center for Aging and Medicine, Huashan Hospital, Fudan University, Shanghai 200040, China; 3National Center for Neurological Disorders, Huashan Hospital, Fudan University, Shanghai 200040, China; 4Department of Biostatistics, School of Public Health, Fudan University, Shanghai 200032, China; 5MOE Frontiers Center for Brain Science, Fudan University, Shanghai 200040, China; 6State Key Laboratory of Medical Neurobiology, Institute of Brain Science, Fudan University, Shanghai 200040, China

**Keywords:** gut microbiome, Alzheimer’s disease, mild cognitive impairment, 16S ribosomal RNA

## Abstract

Altered gut microbiota has been reported in individuals with mild cognitive impairment (MCI) and Alzheimer’s disease (AD). Previous research has suggested that specific bacterial species might be associated with the decline of cognitive function. However, the evidence was insufficient, and the results were inconsistent. To determine whether there is an alteration of gut microbiota in patients with MCI and AD and to investigate its correlation with clinical characteristics, the fecal samples from 94 cognitively normal controls (NC), 125 participants with MCI, and 83 patients with AD were collected and analyzed by 16S ribosomal RNA sequencing. The overall microbial compositions and specific taxa were compared. The clinical relevance was analyzed. There was no significant overall difference in the alpha and beta diversity among the three groups. Patients with AD or MCI had increased bacterial taxa including Erysipelatoclostridiaceae, Erysipelotrichales, Patescibacteria, Saccharimonadales, and Saccharimonadia, compared with NC group (*p* < 0.05), which were positively correlated with APOE 4 carrier status and Clinical Dementia Rating (correlation coefficient: 0.11~0.31, *p* < 0.05), and negatively associated with memory (correlation coefficient: −0.19~−0.16, *p* < 0.01). Our results supported the hypothesis that intestinal microorganisms change in MCI and AD. The alteration in specific taxa correlated closely with clinical manifestations, indicating the potential role in AD pathogenesis.

## 1. Introduction

Alzheimer’s disease (AD) is the most common form of dementia and presents with progressive decline in cognition, behavioral and social skills [1]. It features a continuous pathological process that undergoes: the asymptomatic preclinical stage, mild cognitive impairment (MCI), and AD dementia [2]. Early detection is critical for timely intervention and better prognosis [1,3]. Different hypotheses exist regarding AD pathology, mainly including amyloid-β accumulation in plaques, neurofibrillary tangles formation, and neuroinflammatory process [1,4]. However, the precise pathogenesis remains unclear, and disease-modifying treatment is limited.

Recently, gut dysbiosis and specific microbial-based interventions have been reported in neurodegenerative diseases [5,6,7,8]. Probiotics and dietary therapies including the Mediterranean and ketogenic diets are proposed as one of the most effective prophylactic strategies against cognitive deterioration in Alzheimer’s disease [9,10]. However, findings of previous studies on AD were not consistent or were controversial. For instance, a study found that the fecal Lachnospira genera increased in patients with AD but decreased in MCI in comparison with the cognitively normal controls (NC) [11]. Furthermore, the different fecal taxa of AD compared with NC were inconsistent across studies [4,7,11,12,13,14,15]. The underlying reason might be attributed to the innate heterogeneity of the prodromal stage including MCI and insufficient study sample. Moreover, the association between the clinical manifestation and gut microbiota remained largely unknown. Domain-specific cognition had rarely been examined.

Therefore, we analyzed a cohort of 302 fecal samples using bacterial 16S ribosomal RNA (rRNA) gene sequencing, investigated whether there existed an alteration among NC, MCI, and AD. We also evaluated its clinical relevance by exploring the relationship between specific microbiota changes and clinical characteristics, including cognitive performance, disease severity, apolipoprotein E (APOE) genotype, and activities of daily function. We aim to test the hypothesis that (1) there is an alteration in gut microbiota in participants with MCI or AD; (2) specific taxa gradually change along the AD continuum; and (3) these alterations are correlated with AD clinical manifestations.

## 2. Materials and Methods

### 2.1. Study Participants

We recruited 302 Chinese participants in the present study. A total of 94 NC participants were recruited from the Shanghai Aging Study (SAS) [16] which was a community-based cohort in Shanghai, China, while 125 MCI and 83 AD patients were from the Shanghai Memory Study (SMS) [17] that was established based on the memory clinic of Huashan Hospital.

The participants were included if they were: (1) aged 50 years or older; (2) for MCI or AD, diagnosed based on the 2011 National Institute of Aging and Alzheimer’s Association (NIA-AA) criteria; for NC individuals, no evidence of cognitive deficits was determined by neuropsychological tests; (3) able to cooperate with physical examination, neuropsychological tests, and fecal sample collection.

The general exclusion criteria included: (1) cognitive impairment due to other central nervous system disorders, such as Parkinson’s disease (PD), tumors, or epilepsy; (2) cognitive impairment caused by traumatic brain injury; (3) history of taking antibiotics within three months before fecal sample collection; (4) the use of corticosteroid, immunostimulants, and immunosuppressants; (5) history of major gastrointestinal tract surgery in past 5 years; (6) severe gastrointestinal diseases, such as irritable bowel syndrome, inflammatory bowel disease, which had been reported to influence gut microbiota.

All participants (or their legal guardians) provided written informed consent for their participation in the study, which was approved by the Medical Ethics Committee of Huashan Hospital.

### 2.2. Demographics and Assessment of Covariables

During clinical interviews, demographic and lifestyle characteristics were collected, including age, gender, education, medical history (hypertension, diabetes, apoplexy, digestive system diseases), drinking history, and medications (digestive system drugs, antibiotics, hormone therapy drugs).

### 2.3. Neuropsychological Assessment

A whole battery of neuropsychological assessments including the Mini-Mental Status Examination (MMSE) [18,19], Montreal Cognitive Assessment-Basic (MoCA-B) [20], Auditory Verbal Learning Test (AVLT) [21], Boston Naming Test [22], Trail Making Test (TMT) [23], Symbol Digit Modalities Test (SDMT), Rey–Osterrieth Complex Figure Test (ROCFT), Stroop Color–Word Conflict Test (SCWCT), Verbal Fluency Test (VFT), and Clock Drawing Test (CDT) [24], was administered to the participants. The Mandarin version of all these tests had been validated. The comprehensive neuropsychological tests of each participant were performed by a certified neuropsychological rater within a week of fecal specimen collection. Each patient underwent the neuropsychological tests in the same order.

We extracted raw scores from each test to evaluate five clinically significant cognitive domains, including memory, attention, visuospatial function, language, and executive function. In each domain, the proportion of accurate answers was computed. The Z scores were then computed to guarantee that the participants from the two cohorts were comparable [17].

The Clinical Dementia Rating (CDR) Scale, which covers six cognitive, behavioral, and functional aspects, including memory, orientation, judgment and problem-solving, community affairs, home, and hobby performance, and personal care, was used to determine the severity of cognitive impairment [25].

### 2.4. Sample Collection and DNA Extraction

A blood sample was collected, and DNA was extracted. Genotyping of apolipoprotein E (APOE) was accomplished using the Taqman single-nucleotide polymorphism method. APOE 4-positive was defined as having at least one APOE 4 allele.

The fecal sample collection aseptic containers (QIAGEN, Hilden, Germany) together with detailed user guidance were delivered to the participants before the visit. The participants were then required to collect their fasting fecal samples on the day of the clinical visit and send their samples to the hospital within an hour. Returned samples were aliquoted and stored at −80 °C until analysis. Microbial community genomic DNA was extracted from fecal samples using the E.Z.N.A.^®^ soil DNA Kit (Omega Bio-tek, Norcross, GA, USA) according to the manufacturer’s instructions. The DNA extract was checked on 1% agarose gel, and DNA concentration and purity were determined with NanoDrop one UV-vis spectrophotometer (Thermo Scientific, Waltham, MA, USA) by checking the ratios of 260/280 nm and 260/230 nm, respectively.

### 2.5. PCR Amplification and Illumina MiSeq Sequencing

The hypervariable region V3-V4 of the bacterial 16S rRNA gene was amplified by PCR with primer pairs 338F (5′-ACTCCTACGGGAGGCAGCAG-3′) and 806R (5′-GGACTACHVGGGTWTCTAAT-3′) in the ABI GeneAmp^®^ 9700 PCR thermocycler (ABI, Waltham, MA, USA). The PCR amplification of 16S rRNA gene was performed as follows: initial denaturation at 95 °C for 3 min, followed by 27 cycles of denaturing at 95 °C for 30 s, annealing at 55 °C for 30 s, and extension at 72 °C for 45 s, and single extension at 72 °C for 10 min, and end at 10 °C. The PCR mixtures contained 5 × TransStart FastPfu buffer 4 μL, 2.5 mM dNTPs 2 μL, forward primer (5 μM) 0.8 μL, reverse primer (5 μM) 0.8 μL, TransStart FastPfu DNA Polymerase 0.4 μL, template DNA 10 ng, and finally ddH2O up to 20 μL. PCR reactions were performed in triplicate. The PCR product was extracted from 2% agarose gel and purified using the AxyPrep DNA Gel Extraction Kit (Axygen Biosciences, Union City, CA, USA) according to manufacturer’s instructions and quantified using Quantus™ Fluorometer (Promega, Madison, WI, USA).

Purified amplicons were pooled in equimolar and paired-end sequenced on an Illumina MiSeq PE300 platform (Illumina, San Diego, CA, USA) according to the standard protocols by Majorbio Bio-Pharm Technology Co., Ltd. (Shanghai, China). The raw reads were recorded in the NCBI Sequence Read Archive (SRA) database.

### 2.6. Processing of Sequencing Data

The raw 16S rRNA gene sequencing reads were demultiplexed, quality-filtered by FASTP (version 0.20.0, Shifu Chen et al., Shenzhen Institutes of Advanced Technology, Chinese Academy of Sciences, Shenzhen, China) [26], and merged by FLASH (version 1.2.7, Tanja Magoč et al., Johns Hopkins University School of Medicine, Baltimore, USA) [27] with the following criteria: (i) the 300 bp reads were truncated at any site receiving an average quality score of <20 over a 50 bp sliding window, and the truncated reads shorter than 50 bp were discarded, reads containing ambiguous characters were also discarded; (ii) only overlapping sequences longer than 10 bp were assembled according to their overlapped sequence. The maximum mismatch ratio of overlap region was 0.2. Reads which could not be assembled were discarded; (iii) Samples were distinguished according to the barcode and primers, and the sequence direction was adjusted, exact barcode matching, two nucleotide mismatch in primer matching.

Operational taxonomic units (OTUs) with a 97% similarity cutoff [28,29] were clustered using UPARSE (version 7.1, Robert C Edgar, Independent Investigator, Tiburon, CA, USA) [30], and chimeric sequences were identified and removed. The taxonomy of each OTU representative sequence was analyzed by RDP Classifier (version 2.2, Qiong Wang et al., Center for Microbial Ecology, Michigan State University, East Lansing, USA) [30] against the 16S rRNA database (Silva SSU138.1, https://www.arb-silva.de accessed on 16 March 2022) using a confidence threshold of 0.7.

### 2.7. Alpha and Beta Diversity Analyses

A rarefaction analysis based on Mothur (version 1.21.1, Patrick D Schloss et al., Department of Microbiology and Immunology, University of Michigan, Ann Arbor, USA) [31] was conducted to reveal the diversity indices, including the Chao, ACE, and Shannon diversity indices. The beta diversity analysis was performed using UniFrac [32] to compare the results of the principal component analysis (PCA) using the community ecology package, R-forge (Vegan 2.0 package was used to generate a PCA figure, https://r-forge.r-project.org, accessed on 16 March 2022). Mantel tests were carried out to examine the Spearman’s rank correlation between the environmental factors and the bacterial community similarity using Bray–Curtis distance matrices with 999 permutations, using the vegan package in R. A multivariate analysis of variance (MANOVA) was conducted to further confirm the observed differences. The Spearman’s correlation coefficients were assessed to determine the relationships between microbiota and chemical factors such as signaling molecules. The correlation was considered significant when the absolute value of Spearman’s rank correlation coefficient (Spearman’s r) was >0.6 and statistically significant (*p* < 0.05). All statistical analyses were performed by R stats package. R (pheatmap package) and Cytoscape (http://www.cytoscape.org accessed on 16 March 2022) were applied to visualize the relationships through correlation heatmap and network diagrams respectively. Redundancy analysis (RDA) was employed to explore the relationship between environmental factors and bacterial communities. A one-way analysis of variance (ANOVA) test was performed to assess the statistically significant difference in diversity indices between samples. Differences were considered significant at *p* < 0.05. Venn diagrams were drawn using the online tool “Draw Venn Diagram” (http://bioinformatics.psb.ugent.be/webtools/Venn, accessed on 16 March 2022) to analyze overlapped and unique OTUs during the treatment processes. A one-way permutational analysis of variance (PERMANOVA) was performed using R vegan package to assess the statistically significant effects of treatment processes on bacterial communities.

### 2.8. LEfSe Analysis

To further analyze the specific differences of microbiota, and identify biomarkers for highly dimensional colonic bacteria, we used the linear discriminant analysis effect size (LEfSe) method to perform a comparison among the three groups [33]. Kruskal–Wallis sum-rank test was performed to examine the changes and dissimilarities among classes followed by a logarithmic linear discriminant analysis (LDA) to determine the size effect of each distinctively abundant taxa [34], with the LDA score more than 2.0 and *p*-value less than 0.05.

### 2.9. Statistical Analyses

The Shapiro–Wilk test (test for normality) and Levene test (test for homogeneity of variances) were performed before selecting the appropriate parametric or non-parametric test, respectively. The mean and the standard deviation (SD) were used to describe normally distributed continuous variables, while the median (interquartile range (IQR)) was used to describe the skewed distributed continuous variables. For categorical variables, number (*n*) and frequencies (%) were employed. For continuous and normally distributed data, one-way ANOVA test was performed. Kruskal–Wallis H test and Kruskal–Wallis one-way ANOVA test (K samples) were used to analyze the differences between non-normally distributed continuous variables. The Pearson’s chi-squared test and post-hoc z-tests with Bonferroni corrections were used to tell the differences among categorical variables. Pearson’s correlation was performed to evaluate potential correlations between specific gut microbiota and clinical characteristics. Data were analyzed using IBM SPSS Statistics (version 26.0, IBM, Armonk, NY, USA), and R (version 4.0.2). Figures were visualized by GraphPad Prism (version 7.0.0.3, GraphPad, San Diego, CA, USA).

## 3. Results

### 3.1. Demographic and Clinical Characteristics

The demographic and clinical characteristics of individuals with NC, MCI, and AD, were shown in Table 1. No significant differences were found among the three groups in gender, history of diabetes, stroke, or alcohol intake (all *p* > 0.05). The AD patients were younger than the MCI and NC groups (mean age, 71.8, 75.4, 74.3, respectively, *p* = 0.004). The NC group had a higher level of education than MCI and AD (12.4, 11.3, and 9.9, respectively, *p* < 0.001). When compared with the NC participants, individuals with AD or MCI were more likely to be APOE 4 carriers (NC: MCI: AD = 8%: 33%: 52%, *p* < 0.001). Hypertension was more frequent in the NC group (NC: MCI: AD = 55%: 54%: 33%, *p* = 0.027).

Along with the disease progress from NC, MCI to AD, a gradual worsening in ADL, CDR, MMSE, MoCA, and cognitive performance in various domains (Z_memory, Z_attention, Z_visuospatial, Z_executive, and Z_language) was observed.

### 3.2. The Overall Structure of Gut Microbiota among NC, MCI, and AD

As shown in the Venn diagram in Figure 1A, there were a total of 40,639 OUTs investigated, of which 26,263 were common among the three groups. The overall gut microbial compositions of the three groups were shown in Figure 1B at the phylum level. Similar predominant bacteria were found among NC, MCI, and AD. The common predominant bacteria among groups were Firmicutes, Bacteroidota, and Proteobacteria, followed by Actinobacteriota, Fusobacteriota, and Desulfobacterota (Figure 1B).

### 3.3. Alpha and Beta Diversity in NC, MCI, and AD

The alpha diversity index was calculated by species richness or evenness and was compared among NC, MCI, and AD, using rarefaction curve (Appendix A), Shannon–Wiener index (Appendix A), and species accumulation curves (Appendix A). As shown in Appendix A, the analyses of the species abundance and uniformity were appropriate.

Beta diversity was assessed based on the Bray–Curtis analysis (Figure 2A), unweighted unifrac analysis (Figure 2B), and weighted unifrac analysis (Figure 2C). No statistical difference was found among the three groups or in the pairwise comparisons. In non-metric multidimensional scaling (Nmds) (Figure 2D), principal component analysis (PCA) (Figure 2E), and principal co-ordinates analysis (PCoA) (Figure 2F) using Bray–Curtis distance, there was no significant difference among the three groups (Figure 2D–F).

### 3.4. Differences in Specific Microbiota of NC, MCI, and AD

To further analyze the subtle change in microbiota, LEfSe and LDA were performed. Several taxa showed different abundance in specific between-group comparisons: 15 taxa between NC and MCI (Figure 3A), 29 taxa between NC and AD (Figure 3B), and 26 taxa between MCI and AD (Figure 3C). The Venn diagram demonstrated the commonly changed bacteria between each comparison pair.

As shown in Figure 3A–C, the abundance of genus Actinomycetaceae, Actinomycetales, Atopobiaceae, Saccharimonadaceae, and TM7x significantly increased from NC to cognitively impairment (MCI or AD) but showed no difference from MCI to AD. While the abundance of genus Carnobacteriaceae, Erysipelotrichaceae, Gemella, Gemellaceae, Granulicatella, and Staphylococcales Colidextribacter and Oscillibacter significantly altered from non-dementia to AD. Five taxa including the Erysipelatoclostridiaceae, Erysipelotrichales, Saccharimonadales, Patescibacteria, and Saccharimonadia were found continuously altered from NC to MCI, then to AD (Figure 3D).

### 3.5. Association between Gut Microbiota and Clinical Characteristics

The correlations between the above five taxa (Erysipelatoclostridiaceae, Erysipelotrichales, Saccharimonadales, Patescibacteria, and Saccharimonadia) and clinical characteristics were analyzed (Figure 4). All five taxa were positively associated with the CDR score while negatively associated with Z_memory. The abundance of Saccharimonadales, Patescibacteria, and Saccharimonadia were negatively correlated with MMSE, MoCA, Z_executive, and Z_language, while positively correlated with age, higher CDR, and poorer ADL. The Erysipelatoclostridiaceae was negatively correlated with Z_visuospatial. No significant correlation between these taxa and education or Z_attention was observed. The detailed correlation coefficients and *p*-value are summarized in Appendix A.

### 3.6. Abundance Analysis of Five Specific Taxa

Focusing on these five taxa, including Erysipelatoclostridiaceae, Erysipelotrichales, Saccharimonadales, Patescibacteria, and Saccharimonadia, we further found that the abundance of the taxa showed a remarkable increase from NC to MCI, then to AD (Figure 5A) (all *p* < 0.001). We divided the participants into five subgroups according to the CDR score. Again, these five taxa showed an increasing trend as CDR ascended from 0 to 1. Then, the abundance of genus Erysipelatoclostridiaceae and Erysipelotrichales showed a gradual reduction from CDR = 1 to 3, but the abundance of genera Saccharimonadales, Patescibacteria, and Saccharimonadia continued to increase from CDR 1 to 3 (Figure 5B) (all *p* < 0.001). Similar results were found for the APOE 4 analysis. For all five taxa, the abundance in APOE 4 carriers was significantly higher than the non-carriers (Figure 5C) (all *p* < 0.05). These findings indicated that we had discovered five bacterial communities that were potentially associated with AD pathogenesis, which needed to be further explored. The results of between-subgroup comparisons of all study participants and those aged 60 years or older are presented in Appendix A, respectively.

## 4. Discussion

This study compared gut microbiota in participants with NC, MCI, and AD, which showed similar general structure among the three groups. However, at the genus level, compared with NC, several taxa were found altered in MCI or AD, of which five manifested a gradually increasing trend as the disease progressed. Further analysis revealed a significant correlation between these taxa and the clinical characteristics including age, disease severity, APOE genotype, and global and domain-specific cognition. These findings provide evidence for the involvement of gut dysbiosis in the pathogenesis of AD and highlight its importance for diagnosis and intervention.

Several previous studies have focused on alteration in gut microbiota in the AD continuum, from subjective cognitive decline (SCD), MCI to AD [4,12,13,14,35]. Some reported distinct gut microbiota composition in AD [4,13], while others found altered gut microbiota in the early stages such as SCD [12] or MCI [14]. However, the specific taxonomic differences reported were not consistent. Li et al. investigated 30 NC, 30 MCI, and 30 AD and found that similar intestinal dysbiosis was presented in MCI and AD, with Escherichia increased in both fecal and blood samples [14]. Guo et al. studied 18 NC, 20 MCI, and 18 AD and demonstrated that patients with AD or MCI had increased beta diversity, with decreased Bacteroides, Lachnospira, and Ruminiclostridium 9, and increased Prevotella at the genus level [11]. Yildirim examined 51 NC, 27 MCI, and 47 AD and found a stratified community structure marked primarily by Prevotella and Bacteroides [2]. Sheng et al. recruited NC with Abeta PET scans and demonstrated that phylum Bacteroidetes was significantly enriched while phylum Firmicutes and Deltaproteobacteria were significantly decreased in Abeta-positive NC [36]. In our study, five genera (Actinomycetaceae, Actinomycetales, Atopobiaceae, Saccharimonadaceae, and TM7x) showed a significant increase from NC to MCI/AD, but no difference was found between MCI and AD, suggesting their potential “ignition” role in the early development of the disease. Eight genera (Carnobacteriaceae, Erysipelotrichaceae, Gemella, Gemellaceae, Granulicatella, Staphylococcales, Colidextribacter, and Oscillibacter) showed a significant alteration from NC/MCI to AD, but no difference between NC and MCI was observed, implying an “accelerator” role in the later progress of AD. It is worth noting that five taxa including Erysipelatoclostridiaceae, Erysipelotrichales, Saccharimonadales, Patescibacteria, and Saccharimonadia continued to alter along the disease continuum, from NC to MCI, then to AD. The inconsistent finding across the studies at the genus level might be attributed to two main reasons. First, the heterogeneity of the participants, especially in the prodromal stage including MCI or SCD. The underlying pathology might not necessarily link to AD. Second, the gut microbiota composition could be affected by factors such as dietary patterns, medicine intake, and comorbid diseases. Sufficient sample size is essential to explore the gut-brain relationship robustly.

Increased Erysipelotrichales, Patescibacteria, Saccharimonadales, and Saccharimonadia have been reported in some degenerative diseases and experimental studies [37,38,39,40,41,42]. Lai et al. found that the abundance of Erysipelotrichales was significantly enriched in a neurotoxin model of PD, another neurodegenerative disorder [40]. The genera Saccharimonadales, Patescibacteria, and Saccharimonadia were found to be related to neurobehavioral symptoms [41]. Irina S et al. reported that mitochondrial function was related to changes in gut microbiota, like Patescibacteria, and Saccharimonadia [42]. It was proposed that specific microbial metabolites were sufficient to promote neurodegeneration and disease symptoms. An altered gut microbiome could exacerbate chronic neuroinflammation through fatty acid mediated inflammatory pathways, which eventually mediated amyloid-beta deposition and neurofibrillary tangles formation [37,43]. These results, together with our findings, collectively suggested that specific gut microbes need further mechanism exploration.

Nutrition and dietary interventions have been proposed as viable approaches to improve health and prevent dementia through gut microbiota modulation. Healthy dietary patterns, such as Mediterranean and ketogenic diets, are promising interventions to attenuate cognitive decline in AD [9,10]. The Mediterranean diet, for instance, which is rich in fiber, bioflavonoids, and omega-3 fatty acids has been linked to slowed cognitive decline and reduced risk of AD, possibly by restoration of healthy gut microbiota [10,44]. A pilot randomized controlled trial (RCT), which included 11 MCI and six cognitively normal older adults, reported that a modified Mediterranean ketogenic diet could modulate specific microbiota patterns associated with AD cerebrospinal fluid profile, implying its therapeutic potential [8]. Another 24-month RCT in a European population, involving 612 non-frail or pre-frail participants also verified that the Mediterranean diet could reduce frailty and improve cognition via modifying gut bacteria [45]. Erysipelotrichales, one of the five taxa found in this study that increased constantly from NC, MCI, to AD, was previously shown to be altered by a high-fat diet in rats and to be modulated by a diet containing Tetragonia tetragonioides [37]. More dietary intervention trials are required to verify our findings and explore whether certain dietary patterns might prevent AD through gut microbiota modulation.

Another major concern regarding the gut microbiota study in AD is the clinical relevance. Some studies have preliminarily explored the clinical association of the differentially enriched taxa [4,7,11,13,14]. The SILCODE study showed that alterations in the gut microbial composition were associated with cognition, especially memory [12]. In the current study, comprehensive clinical characteristics were collected, including age, gender, education, APOE genotype, cognitive domains, daily function, and disease severity. The genera Saccharimonadales, Patescibacteria, and Saccharimonadia were positively correlated with age. Erysipelatoclostridiaceae, Erysipelotrichales, Saccharimonadales, Patescibacteria, and Saccharimonadia were positively associated with APOE 4 genotype. APOE 4 was a strong genetic risk factor for AD [46,47,48], with the AD risks increased by 3–4 times and 8–12 times, respectively, for carriers with APOE3/Ε4 or APOΕ4/Ε4 [46]. Tran found an association between the APOE genotype and the gut microbiota in both human and transgenic mice [49]. The results from our study showed that all five specific taxa were differently enriched in APOE 4 carriers compared with the non-carriers. Since APOE is specific to AD and is believed to be involved in the pathogenesis, these findings suggested a potential pathological role of these gut genera. In the correlation analysis with global cognition, various cognitive domains, the severity of dementia, and ability of daily living, the five taxa consistently demonstrated high relevance.

Strengths of this study include: (1) The MCI and AD patients were from a well-established clinical cohort (SMS), whereas the NC individuals were from a long-term regularly-followed community-based cohort (SAS). Diagnosis for each participant was established through a standardized procedure by a consensus group involving neurologists, neuropsychologists, and study nurses. Considering AD and MCI were prevalent among the elder population, all the NC participants enrolled in this study had been followed for at least five years, to ensure the stability of the normal cognitive status and the reliability of the diagnosis. (2) Each individual received a neuropsychological assessment, and more than 85% of them were administered a comprehensive test battery covering global cognition, memory, executive function, attention, visuospatial ability, language, and daily function. This enabled us to perform an in-depth analysis between gut microbiota and clinical phenotypes. To the best of our knowledge, few studies have reported the association with domain-specific cognition previously. (3) The fasting fecal sample was collected and processed according to a standardized procedure to avoid any interference. Comorbid diseases and medications that might influence the gut microbiota were excluded strictly. There were also some limitations of this study: (1) Although a sample of 302 participants was relatively large compared with most previous research, the statistical power could still be inadequate to detect subtle microbiota change, and the normal reference level could not be attained. Moreover, this could be the underlying reason for the insignificant difference in general gut structure among groups. In addition, given the relatively inadequate sample size in our study, it was not possible to control all confounding variables, for instance, the bias of hypertension among groups. For further investigation, we will require additional cohorts to expand the sample size to validate our main findings and further explore the influence of cardiovascular factors in the study of AD and gut microbiota. (2) As a single-center, cross-sectional study, the generalizability of this study was limited. Multicenter research with a longitudinal design and larger sample size is warranted to replicate the results. (3) Patients with advanced dementia were not enrolled because of the difficulty in obtaining the fecal sample, which possibly biased the results towards a negative finding. (4) Information on dietary patterns and nutrition was not available in this study, which prohibited us from exploring the relationship between gut microbiota population and food style. Future investigation will collect more nutrition-related information to explore nutrition and gut microbiota in this memory cohort. (5) AD dementia and MCI in our study were diagnosed mainly based on the clinical criteria rather than pathological evidence. Although we had deepened the analysis by exploring detailed clinical relevance from various aspects, future research incorporating PET scanning or CSF analysis is still warranted to assess the association between gut microbiota and pathological change. (6) Our study suggested the role of the gut–microbiota–brain axis in the development of AD, whereas the precise mechanism still needs to be clarified through functional experiments.

## 5. Conclusions

This study identified significant alteration in some specific gut microbiota in individuals with MCI and AD, and clearly demonstrated their relevance with core clinical features. Our results suggested that gut microbial composition might develop as early as the MCI stage, which provided a potential target for timely diagnosis and effective intervention.

## Figures and Tables

**Figure 1 nutrients-14-03959-f001:**
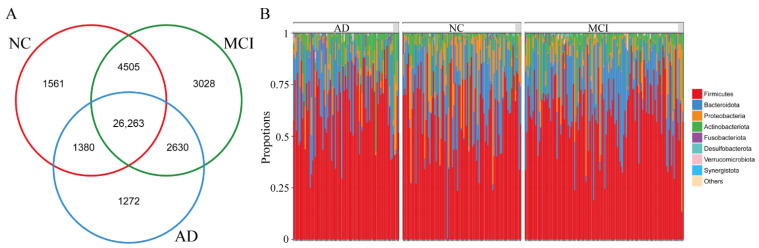
The overall structure of the gut microbiota based on the analysis of microbial diversity among NC, MCI, and AD. (**A**) Venn diagram showing the overlap of the OTUs found in the gut microbiota among NC, MCI, and AD. (**B**) The gut microbial compositions at the phylum levels among NC, MCI, and AD. NC, cognitively normal controls; MCI, mild cognitive impairment; AD, Alzheimer’s disease; OTUs, operational taxonomic units.

**Figure 2 nutrients-14-03959-f002:**
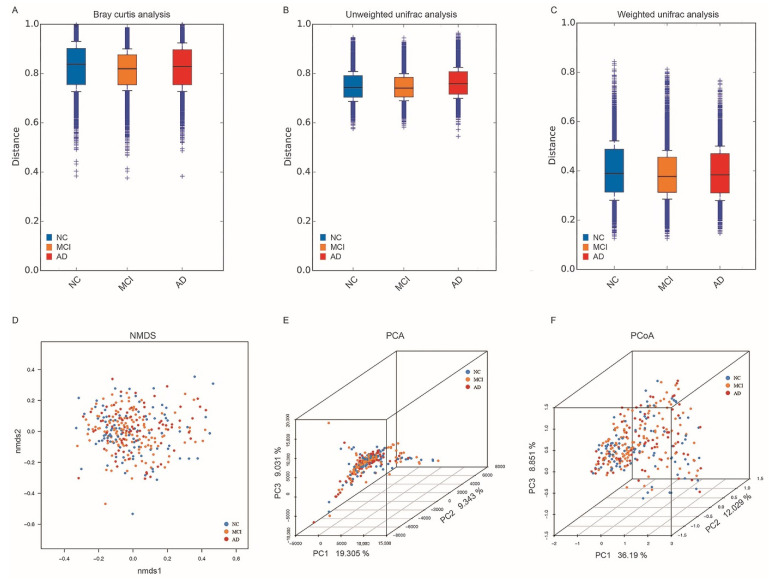
Beta diversity analysis in NC, MCI, and AD. (**A**–**C**) The beta diversity of NC, MCI, and AD by bray–Curtis (**A**), unweighted unifrac (**B**), and weighted unifrac (**C**) analyses. (**D**) Non-metric multidimensional scaling (NMDS) analysis of the gut microbial in NC, MCI, and AD. (**E**) Principal component analysis (PCA) among the three groups. (**F**) Principal co-ordinates analysis (PCoA) among the three groups. NC, cognitively normal controls; MCI, mild cognitive impairment; AD, Alzheimer’s disease; Nmds, non-metric multidimensional scaling; PCA, principal component analysis; PCoA, principal co-ordinates analysis.

**Figure 3 nutrients-14-03959-f003:**
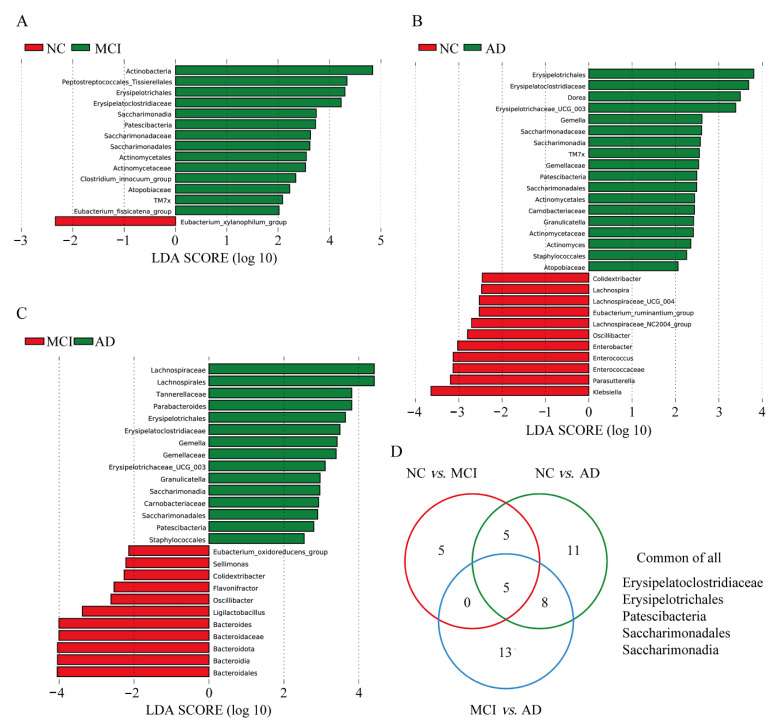
Bacterial taxa with different abundances among NC, MCI, and AD. (**A**) The differences of the LDA scores, histogram for bacterial genera between NC and MCI. (**B**) The differences of the LDA scores, histogram for bacterial genera between NC and AD. (**C**) The differences of the LDA scores, histogram for bacterial genera between MCI and AD. (**D**) Venn diagram of the genera showing the differences among the three groups. NC, cognitively normal controls; MCI, mild cognitive impairment; AD, Alzheimer’s disease; LDA, linear discriminant analysis.

**Figure 4 nutrients-14-03959-f004:**
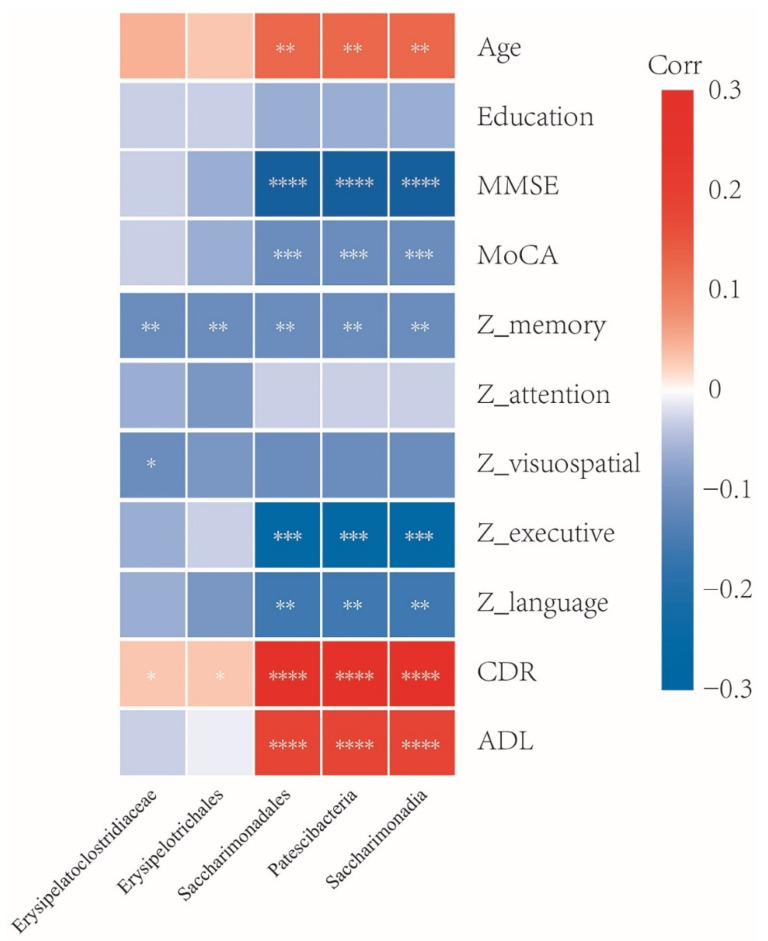
Correlations between the five specific taxa and clinical characteristics. The correlation coefficients (Corr) are displayed. Red or blue signified positive or negative correlation, respectively. MMSE, Mini-mental State Examination; MoCA, Montreal Cognitive Assessment; ADL, Activities of Daily Living; CDR, Clinical Dementia Rating; The composite Z scores were computed for specific cognitive domains including memory, attention, visuospatial ability, language, and executive function. * *p* < 0.05, ** *p* < 0.01, *** *p* < 0.001, **** *p* < 0.0001.

**Figure 5 nutrients-14-03959-f005:**
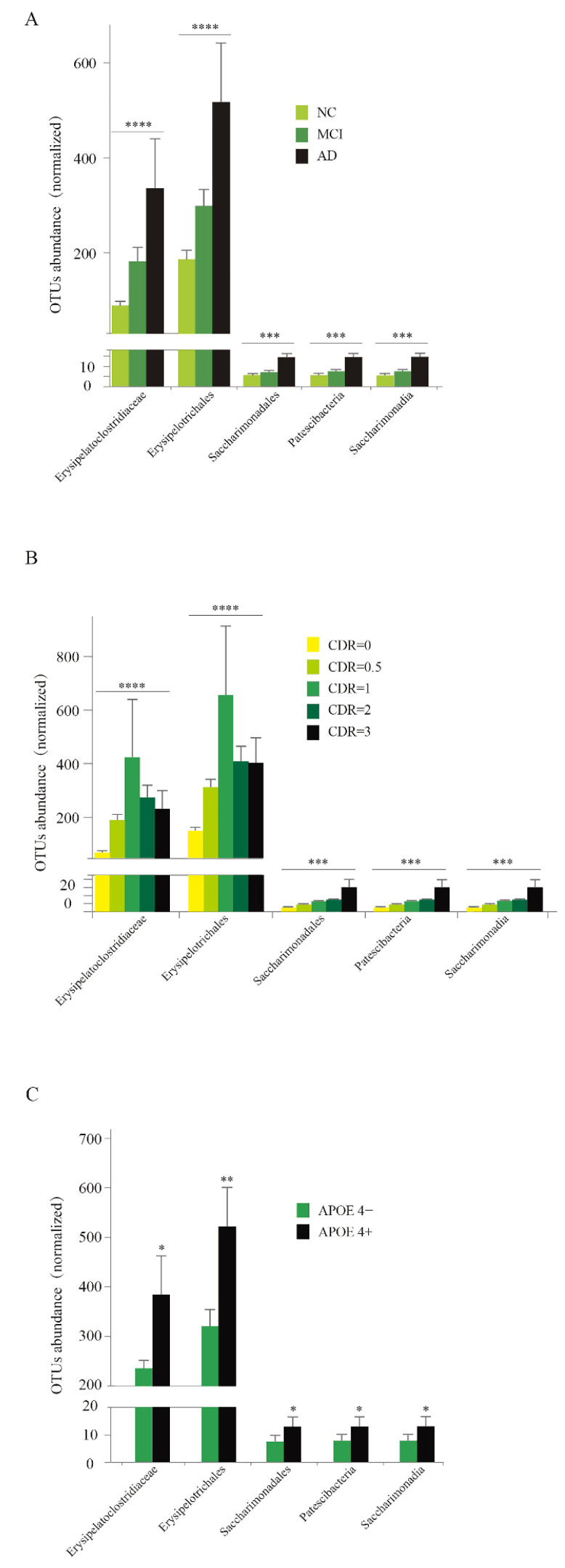
The abundance of the five specific taxa among different clinical subgroups. (**A**) Comparison of the abundance of five taxa Erysipelatoclostridiaceae, Erysipelotrichales, Saccharimonadales, Patescibacteria, and Saccharimonadia in NC, MCI, and AD. (**B**) Comparison of the abundance of five taxa Erysipelatoclostridiaceae, Erysipelotrichales, Saccharimonadales, Patescibacteria, and Saccharimonadia in CDR 0, 0.5, 1, 2, and 3 subgroups; (**C**) Comparison of the abundance of five taxa Erysipelatoclostridiaceae, Erysipelotrichales, Saccharimonadales, Patescibacteria, and Saccharimonadia in APOE 4 positive (APOE+) or negative (APOE−) subgroups. NC, cognitively normal controls; MCI, mild cognitive impairment; AD, Alzheimer’s disease; CDR, Clinical Dementia Rating; APOE, apolipoprotein E; OTUs, operational taxonomic units. * *p* < 0.05, ** *p* < 0.01, *** *p* < 0.001, **** *p* < 0.0001.

**Table 1 nutrients-14-03959-t001:** Demographic and clinical characteristics among study participants.

Characteristics	Total (*n* = 302)	Clinical Diagnosis			*p* Value
		NC (*n* = 94)	MCI (*n* = 125)	AD (*n* = 83)	
Gender, female, *n* (%)	187(61.9)	58(61.7)	76(60.8)	53(63.9)	0.905
Age, yr, mean (SD)	74.1(8.7)	74.3(10.6)	75.4(7.1)	71.8 (8.3) ^&^*	0.004
Education, yr, mean (SD)	11.3(3.9)	12.4(3.8)	11.3(3.6) ^#^	9.9(4.1) ^&^	<0.001
APOE 4 positive, *n* (%)	93(32.7)	8(8.9)	33(29.5) ^#^	52(63.4) ^&^*	<0.001
Hypertension, *n* (%)	142(47.2)	55(58.5)	54(43.2)	33(40.2) ^&^	0.027
SBP, mmHg, median [Q1, Q3]	140.0 [127.8, 152.0]	142.0 [129.0, 153.0]	139.5 [127.0, 152.0]	138.0 [125.0, 152.5]	0.307
DBP, mmHg, median [Q1, Q3]	76.0 [70.0, 83.0]	78.0 [70.0, 84.0]	75.5 [69.0, 83.0]	75.0 [70.5, 81.5]	0.782
Diabetes mellitus, *n* (%)	43(14.3)	13(13.8)	20(16.0)	10(12.2)	0.738
Stroke, *n* (%)	48(15.9)	11(11.7)	25(20.0)	12(14.6)	0.234
Alcohol intake, *n* (%)	39(13.0)	11(11.7)	18(14.5)	10(12.2)	0.803
MMSE score, median [Q1, Q3]	27.0 [23.0, 29.0]	29.0 [28.0, 30.0]	27.0 [26.0, 29.0] ^#^	19.0 [14.0, 22.0] ^&^*	<0.001
MoCA score, median [Q1, Q3]	20.0 [15.0, 24.0]	25.0 [23.0, 27.0]	22.0 [19.0, 24.0] ^#^	12.0 [7.5, 17.0] ^&^*	<0.001
ADL score, median [Q1, Q3]	20.0 [20.0, 22.0]	20.0 [20.0, 21.0]	20.0 [20.0, 21.0]	22.0 [20.0, 31.0] ^&^*	<0.001
CDR score, median [Q1, Q3]	0.5 [0, 1]	0 [0, 0]	0.5 [0.5, 0.5] ^#^	2 [1, 2] ^&^*	<0.001
Z_memory, median [Q1, Q3]	−0.02 [−0.97, 0.79]	0.92 [0.57, 1.5]	−0.22 [−0.81, 0.28] ^#^	−1.31 [−1.31, −0.97] ^&^*	<0.001
Z_attention, median [Q1, Q3]	0.02 [−0.63, 0.69]	0.5 [−0.04, 1.03]	0.03 [−0.45, 0.65] ^#^	−1.02 [−1.62, −0.35] ^&^*	<0.001
Z_visuospatial, median [Q1, Q3]	0.29 [−0.17, 0.61]	0.57 [0.25, 0.72]	0.22 [−0.17, 0.48] ^#^	−0.42 [−1.75, 0.09] ^&^*	<0.001
Z_executive, median [Q1, Q3]	0.25 [−0.08, 0.46]	0.43 [0.26, 0.56]	0.22 [−0.08, 0.43] ^#^	−0.15 [−0.52, 0.05] ^&^*	<0.001
Z_language, median [Q1, Q3]	0.15 [−0.33, 0.63]	0.63 [0.29, 0.91]	0.09 [−0.29, 0.47] ^#^	−0.54 [−1.1, −0.17] ^&^*	<0.001

NC, cognitively normal controls; MCI, mild cognitive impairment; AD, Alzheimer’s disease; APOE, apolipoprotein E; SBP, Systolic blood pressure; DBP, Diastolic blood pressure; MMSE, Mini-mental State Examination; MoCA, Montreal Cognitive Assessment; ADL, Activities of Daily Living; CDR, Clinical Dementia Rating. The composite Z scores were computed for specific cognitive domains including memory, attention, visuospatial ability, language, and executive function. ^#^
*p* < 0.05, comparison between NC and MCI; ^&^
*p* < 0.05, comparison between NC and AD; * *p* < 0.05, comparison between MCI and AD. APOE genotype information was available in 90 NC, 111 MCI, and 82 AD participants.

## Data Availability

Not applicable.

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
