# Peer review of "Altered Gut Microbiota and Its Clinical Relevance in Mild Cognitive Impairment and Alzheimer’s Disease: Shanghai Aging Study and Shanghai Memory Study"

_nutrients, 2022, doi:10.3390/nu14193959_

Round 1
Reviewer 1 Report
The authors show important novel data which contributes for the understanding of neurodegenerative diseases
Abstract
The authors minimized what other have done in previous publications about this issue but these same authors do not close their abstracts with a strong conclusion. The words … which helped further… do not sound clear.
Introduction section.
It would be nice if the authors could formulate the hypothesis to be tested (example: In the present study we test the hypothesis that….
Material and Methods Section
Line 66. Why aged 50 years or older? There is a general agreement about that ? Why not 60 years or older? I am curious to know how is the difference in the main parameters when comparing the following subgroups: a group with patients ranging from 50 to 59 years old versus a group composed by the other older patients.
Why the exclusion criteria did not include those under antibiotics medication
It will be important to say if the numerical data were first of all subjected to analysis of a normal distribution to then decide to do a parametric analysis.
The table 1 does not show important characteristics, such as systolic and diastolic blood pressure, resting heart rate. The cardiovascular system function should have been explored a little better.
Considering the great number of testes performed , did the sequence of each on was int the same order and intervals for all patients? This may be considered and commented in Material & Methods and in the Discussion sections
Considering the limitations of those patients to perform the tasks commonly used to determine, for instance cognitive deficits, was not applied by a unique person. In case this issue occurred, it would be important to consider as a limitation and to discuss possible individual discrepancies.
At the subsection 2.3 the described that they applied a Stroop test. They should explain if that active stressor test was
Results Section
The authors show provided a table of characteristics between different groups. It is important to know about their history of some parameters. However, I can not understand why crucial data such as values of blood pressure and heart rate should be measured because it is well known that the disturbance of cardiovascular function is an important contributor to neurodegenerative disorders, such as dementia. Did you blood pressure ? How you explain a so high percentage of hypertension in the NC (more than 50%). I would suggest the authors to show in that table resting values of systolic/diastolic and also how were counted those exhibiting normotension but due to the action of antihypertensive medication. In this regard, that table shows an inverse condition when they analyzed other chronic diseases such as diabetes.
Gut Microbiota by comparing the abundance
The authors of this manuscript performed the analysis of several parameters, including the comparisons of abundance in the phylum level summing the amount of Firmicutes plus Bacteroidetes. However, the reader also needs to know the percentage that was considered a reference level of normality in of the gut microbiota. Based on the heatmap diagram at the genus (Figure 1C) and phylum level (Figure 1D) of 218 all the participants, similar predominant bacteria were found (Figure 1C and 1D). You ask us to see Figure 1C and 1D, showing the gut microbial compositions at the phylum levels. However, is almost impossible to read what is on, when looking at the Figure 1. It looks like an arial #4.
Please, try to be clearer when interpreting the findings about microbial diversity. It also will be nice if you say which were the 3 or 5 main protective and the 3 or 5 main pathogenic that were observed in dementia patients when compared with control group.
Reviewer 2 Report
The authors demonstrated that the alteration in gut microbiota correlated closely with clinical manifestations, which helped further understand the pathogenesis of Alzheimer’s disease (AD) and explore potential therapeutic targets. The present study was well organized and well investigated, and will give us a new information especially in the field of gut microbiota. To improve the quality of this paper, the authors should revise it according to the following suggestions;
1) The recent many reports have demonstrated the close relationship between gut microbiota population and food style or life style. Especially in Journal "Nutrients", the authors should show the background data of food style or nutrition to understand the role of gut microbiota in mild cognitive impairment (MCI) and Alzheimer’s disease (AD).
Round 2
Reviewer 2 Report
The authors revised it partially. The present study was well organized and well investigated, and will give us informations of the relationship between gut microbiota and MCI and AD. However, the present study did not have any nutritional data.
